# The Experience of Apathy in Dementia: A Qualitative Study

**DOI:** 10.3390/ijerph18063325

**Published:** 2021-03-23

**Authors:** Waqaar Baber, Chern Yi Marybeth Chang, Jennifer Yates, Tom Dening

**Affiliations:** Mental Health and Clinical Neurosciences, School of Medicine, University of Nottingham, Nottingham NG7 2UH, UK; waqaar.baber@nhs.net (W.B.); marybethchang98@gmail.com (C.Y.M.C.); jennifer.yates@nottingham.ac.uk (J.Y.)

**Keywords:** dementia, carers, apathy, motivation, activities, relationships, qualitative, burden

## Abstract

We aimed to explore and gain an understanding into how people with dementia experience apathy, and consequently suggest effective interventions to help them and their carers. Twelve participants (6 dyads of 6 people with dementia and their family carers) were recruited from “memory cafes” (meeting groups for people with dementia and their families), social groups, seminars, and patient and public involvement (PPI) meetings. People with dementia and their carers were interviewed separately and simultaneously. Quantitative data were collected using validated scales for apathy, cognition, anxiety, and depression. The interviews were semi-structured, focusing on the subjective interpretation of apathy and impacts on behaviour, habits, hobbies, relationships, mood, and activities of daily living. Interviews were recorded and transcribed. Transcripts were analysed using interpretative phenomenological analysis (IPA), which generated codes and patterns that were collated into themes. Four major themes were identified, three of which highlighted the challenging aspects of apathy. One described the positive aspects of the individuals’ efforts to overcome apathy and remain connected with the world and people around them. This study is the first to illustrate the subjective experience of apathy in dementia, portraying it as a more complex and active phenomenon than previously assumed. Apathy and its effects warrant more attention from clinicians, researchers, and others involved in dementia care.

## 1. Introduction

Apathy, derived from the Greek *apatheia*, is defined as freedom from, or insensibility to, passion or feeling. Stoic philosophers apparently regarded apathy as the highest condition of humanity [1]. Marin et al. [2] defined apathy as a disorder of motivation and emotionality or the drive to initiate activity. Apathy can be regarded both as a symptom and as a complex neurobehavioural syndrome [3]. It is one of the most common neuropsychiatric features of dementia, affecting 50–70% of people with the condition [4]. Apathy frequently has adverse effects upon carer well-being [5]. For example, carers often find it frustrating when it appears that the person with dementia is capable of doing a task but simply does not bother, or will only do so if the carer uses strong exhortation.

Apathy and depression frequently coexist in dementia, however either can occur without the other, so they are generally regarded as separate entities, a view supported by biomarker evidence [6]. The presence of apathy is associated with less favourable outcomes in dementia, including more rapid functional and cognitive decline [7], greater disease severity [8], and increased mortality [9]. Furthermore, there is a lack of effective pharmacological treatments with clinically meaningful benefits [10,11]. Non-pharmacological approaches may be beneficial, however the evidence is limited by variations in methodology, for example not using apathy as a primary outcome [12].

Most research on apathy in dementia has taken a quantitative or neurobiological approach. Several rating scales have been developed to measure apathy, e.g., the Apathy Evaluation Scale [13]. However, there is a lack of evidence regarding the lived experience of apathy from the perspectives of people with dementia or carers, and indeed there are relatively few studies using qualitative methods at all. This suggests that to date, there are apparently no studies reporting on the subjective experience of having apathy related to dementia, and none which address the question of how it feels to have this symptom. The closest related study that we were able to identify from a literature review [14] was that of Simpson et al. [15], who studied the subjective experience of apathy in Parkinson’s patients, using a semi-structured interview format with seven participants. Using interpretative phenomenological analysis (IPA), they identified three overlapping themes: (1) reduced motivation to consequences of impairment, (2) unacceptability of apathy, and (3) the social context of apathy.

The need to explore this issue further is important, since understanding how the person with dementia experiences their state of apathy could suggest effective interventions to help them and their carers ameliorate the problems caused by the state of apathy.

## 2. Methods

This study was predominantly a qualitative study based on interviews with dyads of people with dementia and family carers. Interpretative phenomenological analysis (IPA) was used, as it involves an idiographic focus, which means that it aims to offer insights into how a given person, in a given context, makes sense of a given phenomenon [16]. Quantitative data were also collected to illustrate in particular the degree of apathy experienced by the participants. This paper reports data from interviews with people living with a diagnosis of dementia. A companion paper will present data from parallel interviews with family carers of these participants (Chang et al., in preparation).

### 2.1. Participants

Using the terminology of Vasileiou et al. [17], the planned sample size was determined according to (1) considerations of the analysis (to what extent would adequate data quality be attained by the chosen method of IPA?) and (2) pragmatic considerations of available interviewer time and resources needed to analyse the data. On these grounds, it was agreed to recruit a sample of six dyads (each containing a person with dementia plus a carer). Participants were recruited from memory cafes (which are meeting groups for people with dementia and their families), social groups, seminars, and Patient and Public Involvement (PPI) meetings.

Inclusion criteria were as follows:

Person with dementia: diagnosis of dementia; clinical diagnosis of Alzheimer’s disease (for greater homogeneity of sample); able to give informed consent to participate; had an available family carer also able to take part. 

Participants were not selected on the basis of having a particular problem with apathy.

Carers: relative of person with dementia; able to give informed consent to take part.

### 2.2. Ethical Approval

Ethical approval was obtained from Nottingham University Faculty of Medicine and Health Sciences Research Ethics Committee (reference: 449-1912).

### 2.3. Procedure

Prior to the interviews, participants were given information regarding what the study involved, were invited to ask questions, and gave written consent for participation. The persons with dementia and the carers were interviewed separately and simultaneously by two researchers (W.B. and J.Y., respectively) in separate rooms to enable both to speak freely about their experiences. In particular, it was considered important that the voice of the person with dementia could be heard separately from the carer, as there is sometimes a tendency for the carer’s perspective to override that of the person with dementia if both are interviewed together [18].

The interviews were semi-structured (see online Appendix A) and focused on the individual’s subjective interpretation of apathy; and the impact of apathy on the interviewee’s behaviour, habits, hobbies, relationships with others, mood, and activities of daily living. The interviews were audio-recorded with consent, each lasting between 40 and 180 min. The interviews were transcribed verbatim.

### 2.4. Quantitative Data

In addition to the interviews, several scales were completed, including the self- and informant-rated subscales of the Apathy Evaluation Scale [2], short Informant Questionnaire on Cognitive Decline in the Elderly (IQCODE) [19], Montreal Cognitive Assessment (MOCA) [20], Cornell Scale for Depression in Dementia (CSSD) [21], and Hospital Anxiety and Depression (HAD) scale [22]. These scales have been previously validated for assessment of anxiety, depression, and cognition in people with dementia.

### 2.5. Data Analysis

The interview data were analysed using interpretative phenomenological analysis (IPA) [23] to characterise the subjective experience of apathy in individuals with dementia. IPA is a qualitative research methodology that takes an exploratory and inductive approach, with a thorough analysis of personal lived experience, what the experience means to the participants, and their perception of it [24]. IPA applies an idiographic approach focusing on the individual and is subjective in nature rather than a formulation of objective accounts. IPA commonly uses semi-structured interviews as the optimum method, aiming for a complete and in-depth account. The analysis uses a “bottom-up”, inductive approach, whereby codes are generated from the data. In health research, IPA can allow a greater understanding of experiences, which in turn can suggest therapeutic measures to influence health behaviour and lifestyles [25].

Themes were identified using an iterative process, as follows.

Each transcript was read individually and scrutinised by the first author (W.B.) to identify “codes” that were relevant to the phenomenon of apathy. Codes were recorded on a spreadsheet in one column and interpretations and preliminary analysis were recorded alongside quotes in another column. This allowed the analysis of each transcript to be collated in one spreadsheet and allowed cross-inquiry of developing themes.

Each interview spreadsheet was presented to the other members of the research group at weekly team meetings. One interview was discussed at a single meeting, allowing comments and refinements to be made. Consensus about theme content and theme titles was achieved in these team meetings.

Quantitative data were collected mainly for illustrative purposes and only descriptive statistics were used.

## 3. Results

Six people with dementia and six carers were interviewed. As it happened, all of the carers were spouses of the individual with dementia, though this was by chance, not a requirement of the study design. Five people with dementia were men and one was a woman. Conversely, there were five female carers and one male carer. All participants were from a white British background and all were living in the community. The ages of participants and scores on rating scales are shown in Table 1.

The participants with dementia were slightly older than their spouses. They had mild to moderate degrees of cognitive impairment, with a range of 9–22 (out of 30) on the MOCA. They reported moderate levels of apathy on the AES, although it may be noted that their spouses gave them higher scores, as is commonly observed in other studies [26,27].

Four superordinate themes were identified, entitled (1) “losing one’s sense of self”; (2) “feeling like a burden”; (3) “hindered by invisible obstacles”; and (4) “what keeps me going”. The superordinate themes and subthemes identified from analysis of the interview data are presented in Table 2, along with illustrative quotes. Figure 1 graphically represents the relationships between these themes and subthemes, and can be regarded as a model of the subjective experience of apathy in dementia.

### 3.1. Losing One’s Sense of Self

Participants described the loss of their sense of identity and juxtaposed perceptions of their current self with how they felt they used to be at an earlier point in their lives. This juxtaposition was conveyed for some participants as a straightforward account of the physical changes that had happened to them, and for others as a more metaphorical explanation, perceiving the change to involve them turning into someone, or something, else entirely. Participant 03 had a very keen insight into his feelings of apathy and provided a great deal of self-reflection during the interview, as if he had been wrestling with the other being that he was turning into for some time.


*It’s like being two people…the other side of me, the other side, this alien thing.*
(participant 03)

For Participant 03, the development of the “alien thing” as an aspect of their self helps to negotiate firstly the incongruence between their past and present sense of self by separating this aspect of themselves out, and secondly it provides something to anchor the feelings of apathy to—a tangible reason for not initiating activities. The participant wants to do things, however something else separate to their own self prevents them from taking any action.


*You want to do something and you have the ability to do it but you are prevented from doing it for some reason.*
(participant 03)

In contrast, participant 02 maintained his sense of self by contextualising his everyday life within his former role of being a police officer. Whilst this participant’s spouse confirmed to the research team that participant 02 was no longer working, he maintained throughout his interview that he was indeed still a serving officer, and discussed his feelings of apathy within this context. He did not consider himself to do many activities at home, and instead referred to feeling apathetic when he was allocated tasks that would be difficult to resolve or gain a successful outcome.


*Well it’s like thinking that they’re giving you a job to do and you’re not particularly interested in doing it because you know it’s a dead-end job.*
(participant 02)

This participant acknowledged his feelings of apathy regarding such tasks, however did not seem to be particularly troubled by them, potentially because he could relate them to something outside of his own self, in this case his career. A similar process occurred for participant 01, who contextualised her feelings of apathy in terms of constant back pain that had been troubling her for a long time.


*It’s difficult to explain. I get terrible backache and I feel then that I’ve got to try and sit down or take some pills or I don’t know what to do with myself.*
(participant 02)

This participant did not explicitly state that she felt apathetic, but referred to a lack of motivation and no longer doing various activities or tasks that she typically would have done, and always related this to her back problems. Relying on a situation beyond her control helped her to maintain a sense of identity.

Hobbies and activities were important markers of participants’ sense of self. Participants recognised that difficulties in motivating themselves to engage with these pastimes represented another facet of their identity that was slipping away. These changes in sense of self were attributed to two processes. Firstly, participants reported a loss of motivation to initiate or continue doing things that previously were important to them, particularly when left to their own devices. Participants acknowledged that typically they just couldn’t be bothered to do things even when they knew they would enjoy it, and also reported needing their spouses to motivate them with daily activities such as personal hygiene, even though they knew this was important and necessary.


*My granddaughter, she wants me to take her out on the bike. I can’t do that. I don’t want to go places.*
(participant 05)


*The wife pushes me… A push to change clothes. A push to stay tidy and clean, that sort of thing.*
(participant 04)

Secondly, participants spoke of a loss of confidence and a decrease in their self-esteem. This led to feelings of being no longer capable to pursue hobbies or contribute to things around the house. The comparison of a keen engagement in a wide range of past activities, such as hobbies, interactions with other people, and chores, while a current lack of desire to even try to do things demonstrated a shift in identity from someone who is actively involved to someone who is stood on the sidelines of their life. Several participants associated these changes in ability with physical issues, such as back problems (participant 01), weight (participant 05), or their increasing age (participant 04), and this may be interpreted as a way of retaining their sense of self as much as possible by excusing their loss of willingness to try with tangible changes that were beyond their control, as if this was somehow more acceptable and absolved them of having to confront the contribution of their apathy to the change in sense of self.


*I just got old really, I got—not capable of doing it.*
(participant 04)

The changes in sense of self can be interpreted as being driven by what may be considered as the ingredients of apathy, namely losses of motivation and confidence; however, this process also represents a vicious cycle, whereby the apathy further saps the participants’ identity by widening the gap between who they perceived they were in the past and who they perceive themselves to be now. It seems as though once this process has started and the sense of self begins to slip away, there is a feeling of futility towards recapturing it, as though once tarnished there is no value in maintaining it.

### 3.2. Feeling Like a Burden

Apathy manifested itself for most participants as both a contributor and a result of feeling they were a burden upon other people, particularly their spouses or important social groups, and this was driven by a fear of failure. For most, the perception that they would not be able to complete a task to a particular standard, whether the standard they could previously achieve or a standard required by someone else, meant that there was little point in attempting the task, as the likelihood of it going wrong was high.

Sometimes this was based on experience, however often this was based simply on the idea that it probably would go wrong. As participants gradually withdrew from activities and interactions over a period of time, the feeling grew that they would fail and cause someone else an inconvenience because of their failure. Eventually, this feeling was impossible to overcome and represented a block to initiating activities. Participant 03 was previously very active in their church but worried about being able to contribute correctly, and so preferred to reduce their assistance. Unfortunately, this was met with a lack of understanding from others at the church, and the result was further feelings of failure and being a burden.


*I didn’t think it was appropriate for me to carry on because rightly or wrongly I thought I would be more of a hindrance because of memory and motivation and that. And one of the church officials said, I think you are just milking it and that has really, really burnt into my brain, every time I see that person that pops into my brain, that’s really upset me.*
(participant 03)

Participant 05 wanted to help around the house and to make life easier for his wife, who was still working as a paid carer, but struggled to make it happen.


*I feel like I want to help (spouse) but I can’t, and I feel like she thinks—she’s working all the hours she can get and when she comes home, I haven’t done things. She’ll give me tasks to do. She writes it all down and I just haven’t done it or forget to do it.*
(participant 05)

This participant did not expand on what he supposed his wife thought about his lack of action, but the further data suggest that the result is friction in their relationship, and the participant worried he would lose his wife as a result of these instances.


*(Spouse) has to push me, yes. She’ll push me and say, it has to be done. We have a little bit of an argument and I know she’s been to work, she’s done 24 hours work for this person to die, type thing. I know that has to be done. So, I do—I struggle just to get it done but I get it done to keep her. I try and put the washing in. She’ll shout at me, I’ll do the washing. Well I say, you want me to do things. I try to do that, she says no you’ll mix all the colours up.*
(participant 05)

Participant 01 related her changing ability to contribute to the household to her back problems, and did feel a sense of unease at her husband having to take on more tasks. This participant reported several times that her husband was very good towards taking on such tasks and this seemed to relieve some feelings of burden.


*When my back is hurting, I don’t—I feel I can’t do anything at all. Then that worries me then that I’m leaving it to (spouse). I think it’s not fair.*
(participant 01)

To assuage such feelings, participants often engaged in tactics to avoid situations and tasks, because in removing themselves from the possibility of issues occurring, feelings of being a burden may be mitigated. For participant 05, as soon as he sensed that tasks at their allotment were starting to create a sense of burden for his wife, he gave up the allotments.


*(Spouse) got the allotment for me, and then we started having chickens and ducks. We had—there were times I didn’t want to go down and she said come on, we’ll go down. Went down every day. Then the trouble started and I didn’t want to do it anymore. I just got rid.*
(participant 05)

For participant 03, not mentioning ideas to others or reminding them of tasks that he previously undertook reduced the likelihood that he would be sought out to do something, and as such he could avoid creating expectations that might be doomed to failure and result in an inconvenience to others.


*One of the people in there wants to put on a big display of the history of the church and I have most of the information because I gleaned it, I used to be very interested in history, I don’t want to be involved. And yet a year ago, perhaps less than a year ago, I would have been looking forward to doing that but now I am talking myself into, if I don’t mention it to them they may not want the display. If I was to go to them and say do you want me to put on a display again chances are they would say yeah, that would be a good idea.*
(participant 03)

This participant hopes that by keeping quiet, such activities will go away and be forgotten about, effectively getting them off the hook.


*I don’t want to do it, if I don’t mention it they might not mention it and they might go and do something else.*
(participant 03)

Contrastingly, participant 02 did not report avoiding activities or tasks, and this was perhaps related to the participant’s contextualisation of their everyday life within their role as a police officer, where the hierarchy of this particular career did not enable individuals to pick and choose whether they did particular tasks. Instead, if an order to do something was given, a police officer would follow it up with the appropriate action.


*Because then the hierarchy got involved in it…The hierarchy. The bosses.*
(participant 02)

Accompanying the feelings of failure and efforts to avoid ending up in situations that could become problematic for themselves and others, most participants in this sample offered a fairly pessimistic outlook. Participants worried that as their dementia progressed, they would struggle more and contribute less to their households and social roles. Such feelings of pessimism and the potential burden created as their abilities changed in the future fueled the cycle of apathy, whereby the uncertainty of knowing what might happen in any given situation or the possibility that they could create a problem because of their cognitive changes prevented them from engaging in tasks or interactions that they otherwise would have done.


*I think from my point of view with the problems I’ve got, and those problems are getting worse, when it’s somebody who is in need, I could actually make it worse because it’s so unpredictable Alzheimer’s, I don’t know what’s going on tomorrow and something terrible might happen tomorrow.*
(participant 03)

### 3.3. Hindered by Invisible Obstacles

On the surface, apathy can look like there is very little activity taking place, and people who are described as apathetic are generally considered to be unengaged in any particular action or interaction with the environment and people around them. However, the participants in this sample revealed that there is actually a great deal of activity occurring, in that a constant background struggle was taking place between a yearning to do things and a variety of challenges to prevent them from doing things.


*The apathy is your inability to do something about something, you know what you want to do, you know how to do it but you are not capable of actually initiating it.*
(participant 03)

For some participants, it was not clear what the challenges were other than they just generally did not want to do things, or for reasons that they did not know they just could not or would not do them.


*I do get a bit irritated at times because things are untidy, but then, as you see, my wife has got arthritis so she’s a bit restricted, things get dropped and it irritates me that no one is picking them up, but then why don’t I pick them up? I often don’t.*
(participant 03)

Participant 03 described situations where any form of disruption or deviation from the plan would cause any activities to come to a sudden halt, and that picking the activity back up from where they left off and continuing with it became impossible once this had happened.


*About a month ago I think it was, the first thing that I was going to do was, no the second thing, the second, third thing, was the bank, but when I got there they weren’t open because they opened, it must have been on a Wednesday, I couldn’t go to the bank. And the whole route routine then fell apart because I couldn’t do it.*
(participant 03)

For this participant, there is a reliance on momentum, which may be interpreted as being in a state of flow once an activity or sequence of events has begun, where the engagement in such tasks carries them along and protects them from potentially overthinking what they are doing before the feelings of apathy can creep in and stop them. However, if the flow of engagement is broken for any reason, the activity is derailed, and trying to get back on track is too hard. The invisible struggle starts again, and typically the apathy wins.

Participants also described stigma surrounding their apathy and they felt that it was difficult to explain to others, because other people did not understand how they felt or the invisible struggle that they were dealing with.


*You can’t really relate to it or relate it to your own family, because they know the situation, they are part of it so you can’t have a conversation really about how you are feeling with your own family.*
(participant 03)

### 3.4. What Keeps Me Going

Whilst most participants presented a relatively bleak picture, participant 06 in particular remained positive and provided several ideas regarding ways to overcome feelings of apathy. Firstly, this participant spoke of being motivated by wanting to help other people, as this could spur him on to overcome feelings of apathy, even toward jobs that were not particularly enjoyable (see Table 2). This was also suggested as a facilitator by participant 03 and potentially the thought of letting someone down when they are expecting assistance is enough of a trigger to galvanise people into action and actually bring a positive sense of anticipation of a task.


*I looked forward to doing that with him because there are two people. He wants my help to do this, he needs my help to do this, well he doesn’t need it, that’s where the motivation part is coming in. When somebody else is involved and they want to do it, it gives some sort of impetus to overcome the apathy.*
(participant 03)

Having other people involved in activities and doing tasks with others also seemed to be a helpful strategy, particularly if those involved were younger generations and the activity involved an element of teaching or passing skills, for example to their grandchildren. Participant 03 reflects that he is motivated to engage in wood carving when he is teaching his granddaughter how to do it, but would not do this activity only for himself or by himself. Potentially just being around grandchildren and involved in their worlds has a positive impact, as though their energy transfers to the participants and provides them with a reason to act. Consequently, even though grandchildren could be considered quite hard work, they were very much appreciated.


*I still love to have the grandkids.*
(participant 05)

Maintaining hobbies from across the participants’ lives was also thought to be beneficial in overcoming apathy, and activities that participants were particularly invested in or that they were very well practised at seemed to bypass the invisible struggle.


*I go out and feed my fish, which is in the back yard. I’ve got a big fish-pond-type thing.*
(participant 05)

Encouraging a sense of connection to such hobbies can also help to mitigate a loss in the sense of self, as discussed in the first theme, whereby some participants in our sample struggled to continue with hobbies and activities that were a part of who they were. Extra support and encouragement to do this may be particularly helpful.

Lastly, participants in our sample spoke of the importance of routine. Potentially, an established routine left no gaps for the apathy to fill and a defined sequence of events meant that there was no requirement for any decisions about whether to do a task or not. Instead, participants just knew that a particular action came next and they were able to do it without doubting themselves or questioning the situation.


*Normally we will go shopping, into town for coffee, walk around (country) park, see friends.*

*Get dressed, the wife shaves me. Watch the news for a couple of hours, and then we go shopping, come back, and then go out perhaps in the afternoon to parks and gardens or whatever.*
(participant 04)

Removing the element of thought or decision-making helped participants because it reduced the cognitive burden. Making decisions about what to do is not easy, and potentially apathy is related to decision fatigue, where constantly having to think about different potential options and select which one to do is overwhelming and too much for people with dementia to manage in the context of changing levels of cognitive functioning. Participant 05 speaks of his experiences of becoming overwhelmed and needing to remove himself from particular situations to clear his head.


*It annoys me and I’m quite sad, but I don’t know why. I just would rather go somewhere else sometimes or have a cry. I’ll go to the toilet and just sit there.*
(participant 05)

Consequently, anything that can help to reduce the feelings of being overwhelmed with choices or information was considered helpful in warding off the feelings of apathy that could lead participants to give up before they had even started.

### 3.5. Summary of Themes

The analysis of the descriptions and reflections provided by the participants in this sample of people with dementia gives a detailed insight into how apathy can begin and then envelope individuals, chipping away at their sense of self until they have lost confidence in their ability to engage in tasks and activities. Worries about failing can lead to avoidance tactics and pessimism about the future, ultimately increasing feelings of being a burden upon others. The apathy could be considered a spiral or a trap that is difficult to escape from, and contrary to outsiders’ perceptions, apathy actually involves a lot of cognitive effort, whereby in order to overcome the apathy a struggle against unseen forces must be won. However, the insights from this sample suggest there are ways to help, in the form of support to retain hobbies, establishing routines to bypass the cognitive struggle, and involving other people.

## 4. Discussion

This study is the first to portray the subjective experience of apathy by people living with a diagnosis of dementia. The findings identified four major themes, three of which touched upon challenging aspects of apathy, but one described positive aspects of the individuals’ efforts to overcome apathy and remain connected with the world and people around them.

Prior to the study, we had wondered if interviews with people with dementia would yield sufficient material for analysis, however these concerns were unfounded, as evidenced by the richness of comments in the interview transcripts. It may be assumed that apathy is a passive state with an apparently inert behavioural profile, however these findings show that this is not the case. What emerged was a picture of a struggle against cognitive difficulties, fear of failure, and invisible obstacles, with individuals desperate to remain connected and appreciative of their carers’ efforts to help them, even though clearly they have difficulty in demonstrating such appreciation. Apathy is, therefore, a complex and multifaceted phenomenon. This more active conceptualisation of apathy, as shown in our model in Figure 1, is novel and offers more potential for psychological and social mediation than does the traditional view of apathy as being simply negative and hopeless.

The strengths of this study include using an idiographic method of analysis (IPA), which allows people’s experiences to emerge from the interview data. Four researchers were involved in discussion of each individual interview, and the themes were refined iteratively and agreed by the whole team, thus enhancing the robustness of the analysis and our confidence in the findings. Several versions of the model were also discussed before arriving at a final version.

It might be suggested that the relatively small number of participants (N = 6) and their relative homogeneity of diagnosis, stage of dementia, and ethnic background could be limitations. Certainly, it would require further research to ascertain whether the same themes are relevant in other social and cultural groups. However, IPA is an idiographic method, and the aim of the study was to build a coherent picture of apathy in this sample, for which purpose the sample size was certainly adequate to attain sufficient data quality for the proposed analysis.

A potential criticism of IPA and other qualitative methods of analysis is that they may impose the researchers’ views upon the interpretation on the themes derived from the data [28]. This can occur, for example, if the interview schedule has too much structure and not enough scope for open-ended responses. This study aimed to minimise these shortcomings by using a semi-structured interview format, encouraging open responses, checking that emergent themes did not simply reproduce the interview schedule, and from the input of all four researchers throughout the process of analysis.

The study has implications for further research, for example exploring whether similar themes prevail in different cultural settings, for example with people from Black and Asian minority ethnic groups. There is already evidence to suggest that psychosocial interventions targeted at apathy may be effective [12,29], but clearly more is required in order to support people living with dementia and their family carers. Our model of apathy suggests a means by which researchers can look at the mechanisms by which such interventions may work.

## 5. Conclusions

Apathy can be considered an active struggle rather than a passive state, whereby individuals experiencing apathy yearn to stay connected to, and involved in, the world around them. Various strategies employed by carers are helpful and appreciated. This research needs to be conducted more widely with underrepresented groups, but currently our findings suggest sensible opportunities for intervention that should be explored further. Probably the most important contribution of this study, however, is to challenge the current prevailing view of apathy and to portray it as a more complex and more active phenomenon than is usually assumed. We suggest that clinicians and providers of services for people with dementia pay more attention to apathy and how its effects may be alleviated. There may be a need for facilities that address issues around apathy—perhaps even “apathy cafes”.

## Figures and Tables

**Figure 1 ijerph-18-03325-f001:**
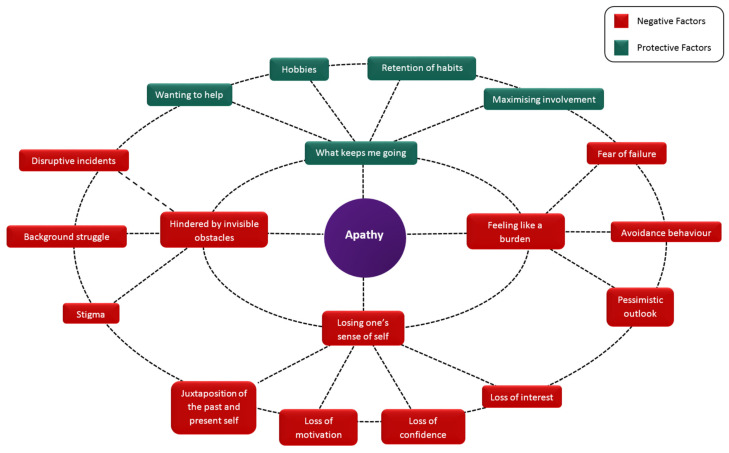
The subjective experience of apathy in dementia, with superordinate themes (inner ring) and subthemes (outer ring).

**Table 1 ijerph-18-03325-t001:** Summary of participant demographic and clinical data.

Participant Characteristics	Dyad 1	Dyad 2	Dyad 3	Dyad 4	Dyad 5	Dyad 6	Mean (SD)
PwD age	84	76	74	72	60	73	73.2 (7.1)
PwD gender	F	M	M	M	M	M	-
Carer age	84	70	68	72	59	63	69.3 (7.9)
Carer gender	M	F	F	F	F	F	-
MOCA	11	9	18	22	18	15	15.5 (4.4)
* Short IQCODE	4.69	4.81	4.4	4.7	4.75	3.88	4.54 (0.32)
AES-Self	38	47	46	44	38	26	39.8 (7.1)
* AES-Informant	47	62	39	65	43	51	51.2 (9.5)
HAD Anxiety	0	2	16	8	14	2	7 (6.2)
HAD Depression	1	3	12	8	13	0	6.2 (5.1)
CSDD—self	1	1	15	3	26	8	9.0 (9.0)
* CSDD—informant	8	11	17	4	26	4	11.7 (7.8)

Note: * = rated by carer, other ratings directly from person with dementia. PwD = person with dementia; MOCA = Montreal Cognitive Assessment (score range 0–30; scores <26 indicate impairment); IQCODE = short Informant Questionnaire on Cognitive Decline in the Elderly (score range 1–5; a score of 3 indicates “no change”, 4 indicates “a bit worse”, and 5 indicates “much worse”); AES = Apathy Evaluation Scale (score range 18–72; lower scores indicate more apathy); HAD = Hospital Anxiety and Depression scale (score ranges of 0–21 for anxiety and 0–21 for depression; a score on either subscale of 8–10 indicates a borderline case, 11–21 indicates a case); CSDD = Cornell Scale of Depression in Dementia (score range 0–38; scores >10 indicate a probable major depressive episode and scores >18 indicate a definite major depressive episode).

**Table 2 ijerph-18-03325-t002:** Superordinate themes, subthemes, and illustrative quotes.

Scheme	Subthemes	Illustrative Quotes
Losing one’s sense of self	Juxtaposition of the past and present self	“I used to be quite a powerful bloke, quite muscly. I’ve never took drugs or anything like that. I was able to lift quite a lot during the army and stuff like that. I was very powerful, but now I haven’t got the strength to lift anything. It’s like—I used to do a lot of boxing and stuff like that, but now I’m not” (Participant 05)
	Loss of interest in hobbies and activities	“I mean I am very conscious of the fact that I just cannot motivate myself to get involved in my hobbies” (Participant 03)
	Loss of motivation	‘’I used to love doing the job I did but then I started getting—I couldn’t be bothered going to it’’ (Participant 05)
	Loss of confidence	“I’m not worthy. I can’t do what I wanted to do…So, I feel like a failure. So, I just don’t bother” (Participant 05)
Feeling like a burden	Fear of failure	“Now I don’t want to be involved in other people’s problems…I am afraid to get into other people’s problems in case I get it wrong.” (Participant 03)“Occasionally—I think I’ve let the wife down for one reason or another” (Participant 04)
	Avoidance Behaviour	“I can do family meetings but I wouldn’t bother about anything that’s going on in the neighbourhood. If I thought about it, I couldn’t do anything about it so I try not to think about it, otherwise I’m worrying” (Participant 01)
	Pessimism	“I’m just scared what comes out of it, what the outcome nobody tells you or that type of thing” (Participant 05)
Hindered by invisible obstacles	Background struggle	“A swan going along a lake, all serene, lovely, but underneath his feet are in complete confusion flapping around just to keep it afloat” (Participant 03)
	Disruptive incidents	“I had started to do a painting and the telephone rang, I wouldn’t know what to do. I wouldn’t know whether to stop the painting and answer the phone, if I answered the phone I forget where I was on the painting and I just wouldn’t carry on” (Participant 03)
	Stigma	“People tend to back away, they don’t want to talk to me and I think that’s not necessarily a bad reaction from me, it’s ignorance, they don’t understand it and they get the wrong impression” (Participant 03)
What keeps me going	Desire to help	“Yes, if I can help somebody, I will help somebody and I’d do that even if I didn’t like the job I was doing but it was helping somebody” (Participant 06)
	Maximising involvement	“I tried to teach my granddaughter how to do wood carving, that was great when I was trying to show her how to do it but I wouldn’t get up and do it myself” (Participant 04)
	Hobbies	“Most of my activities are doing jigsaws….because you get lost with the brain on it and if I get back ache and that then I can leave it for a bit and then lie there and then get back up again and start again” (Participant 01)
	Retention of habits	“If it’s something that is part of my life routine then I do them, I will probably do them without any forethought, I don’t have to think I need to go and get my breakfast specifically, it might take me longer some days than others because I just can’t be bothered to get out the chair, but they will get done” (Participant 03)

## Data Availability

The data presented in this study are available on request from the corresponding author. The data are not publicly available due to participant confidentiality.

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
