# Peer review of "The Experience of Apathy in Dementia: A Qualitative Study"

_ijerph, 2021, doi:10.3390/ijerph18063325_

Round 1

Reviewer 1 Report

The authors have included the range of possible scores below Table 1 as I suggested. However, they have not followed my suggestion to also include in formation on the implication of high or low scores.  I suggest for each of the scales something like:  (score range xx - yy; higher scores = more anxiety).

I have no more suggestions.

Reviewer 2 Report

Thank you for recommending me as a reviewer. This qualitative study was aim to explore and gain an understanding into how people with dementia experience apathy, and consequently suggest effective interventions to help them and their carers. If the authors complete minor revisions, the quality of the study will be further improved.

  1. line 42-51: The introductory section is well written. However, the content of the theoretical background for conducting qualitative research is insufficient. If the author describes the trend and necessity of qualitative research more specifically, it can help readers understand.

2. line 65-78: The procedure and data analysis is well presented. However, the contents of the subject's characteristics are insufficient. If the author describes the subject's characteristics and recruitment procedure more specifically, it can help readers understand.

3. line 129: "Six people with dementia and six carers were interviewed. " - Are the dementia patients and their caregivers a family member (living together)? Also, does carer mean primary caregiver? The author needs to make a more specific definition of carer. Authors need to be more specific about these points in the 'Methods' section or the'Results' section.

Reviewer 3 Report

My overall opinion is that the authors did a good job, reviewing the article according to the suggestions. It is a dimension that has been little worked on in people with dementia, with the potential for providing care that will delay the decline and promote comfort and the ability to interact.

Author Response

This manuscript is a resubmission of an earlier submission. The following is a list of the peer review reports and author responses from that submission.

Round 1

Reviewer 1 Report

This is a very well written and interesting report on the perception and mechanisms of apathy in dementia as experienced by six patients with dementia. Although the number of patients studied is small, the authors make appropriate reservations with regard to the generalization of their observations.

The study will be of interest to a wider group of readers than those familiar with the scales applied for quantitative data.  I therefore suggest that the authors should include for each scale the range of possible scores and a mention of what high or low scores imply.

I have no more suggestions to make

Reviewer 2 Report

Dear authors 

I send you a text with some suggestions in order to improve your text, please read it. 

Good work. 

Reviewer 3 Report

Dear authors,

I presumed some key point when I was reading the abstract in the reviewing proposal. That was what catch my interest on it.

After the revision many of my previous ideas and questions has been solved and I want to congratulate the authors for this work. It is clear, direct, and humble manuscript that contains a great aim behind.

Just for a comment, in some cases anosognosia and apathy is highly correlate although the expressed opinions do not. I understand the use of semi-structured interview could reduce the impact of anosognosia partially but, I would appreciate to find some reference about this phenomenon.

Reviewer 4 Report

Thanks for recommending me as a reviewer. This paper aimed to explore and gain an understanding into how people with dementia experience apathy, and consequently suggest effective interventions to help them and their carers. Twelve participants (6 dyads of 6 people with dementia and their family carers) were recruited from memory cafés, social groups, seminars and Patient and Public Involvement (PPI) meetings. Apathy and its effects warrant more attention from clinicians, researchers and others involved in dementia care. If the authors complete minor revisions, the quality of this study will be further improved.

  1. Abstract: "12 participants (6 dyads of 6 people with dementia and their family carers) were recruited from memory 9 cafés, social groups, seminars and Patient and Public Involvement (PPI) meetings" -> Twelve participants (6 dyads of 6 people with dementia and their family carers) were recruited from memory cafés, social groups, seminars and Patient and Public Involvement (PPI) meetings.

2. line 50-55: This study used interpretative phenomenological analysis (IPA). If the author explains the IPA more specifically, it can help readers understand.

3. line 65-78: Participants are essential in qualitative research. Authors should be more specific description about the participant characteristics.

4. In the case of qualitative research, analysis programs(software) are often used. Didn't the authors use an analysis program in this study?

5. The results of this paper are too verbose. Authors need to separate the codes and tabulate the content of the results.

6. In this study, the conclusion section was omitted. Qualitative research offers readers a variety of implications. The authors need to add the key conclusions of this study.

Reviewer 5 Report

This is an interesting, weel written paper. The authors presented a model of apathy that is relevant for those that are interesting in dementia issues and support to the caregiver, showing mechanisms and building an overall perspective.

Some suggestions

(1) The authors can find the topic "apathy" in studies related to cognitive and functional decline in people with dementia. Moreover, there are some studies related to selfcare dependence that are focused on apathy. My suggestion is to introduce some reflections about selfcare in the introduction;

(2 The informed consent was given also by the people with dementia? Are they able to? And if not?

(3) In the discussion section, there is a need to introduce evidences from studies related to dependence in selfcare in people with dementia